# Chronic Lymphocytic Leukemia (CLL) with Borderline Immunoglobulin Heavy Chain Mutational Status, a Rare Subgroup of CLL with Variable Disease Course

**DOI:** 10.3390/cancers16061095

**Published:** 2024-03-08

**Authors:** Francesco Angotzi, Alessandro Cellini, Valeria Ruocco, Chiara Adele Cavarretta, Ivan Zatta, Andrea Serafin, Stefano Pravato, Elisa Pagnin, Laura Bonaldi, Federica Frezzato, Monica Facco, Francesco Piazza, Livio Trentin, Andrea Visentin

**Affiliations:** 1Hematology Unit, Department of Medicine, University of Padova, 35128 Padua, Italy; francesco.angotzi@studenti.unipd.it (F.A.); alessandro.cellini@studenti.unipd.it (A.C.); valeria.ruocco@studenti.unipd.it (V.R.); chiaraadele.cavarretta@studenti.unipd.it (C.A.C.); ivan.zatta@studenti.unipd.it (I.Z.); andrea.serafin@studenti.unipd.it (A.S.); stefano.pravato@aopd.veneto.it (S.P.); elisa.pagnin@unipd.it (E.P.); federica.frezzato@unidp.it (F.F.); monica.facco@unipd.it (M.F.); francesco.piazza@unipd.it (F.P.); andrea.visentin@unipd.it (A.V.); 2Immunology and Molecular Oncology Unit, Veneto Institute of Oncology, IOV-IRCCS, 35128 Padua, Italy; laura.bonaldi@iov.veneto.it

**Keywords:** chronic lymphocytic leukemia, IGHV mutational status, borderline mutated

## Abstract

**Simple Summary:**

In this study, we present a retrospective analysis of a large cohort of CLL patients with a particular focus on those with a borderline immunoglobulin heavy variable chain mutational status (BL-IGHV) and their disease course. BL-IGHV patients represent a relatively unexplored small fraction of CLL patients whose prognosis is still largely debated. Our work aims to provide further evidence regarding BL-IGHV CLL. The key findings of our research highlight differences in time to first treatment (TTFT) and overall survival (OS) between our cohort of borderline mutated patients and previous experiences, but also the relative similarities shared by BL-IGHV patients across the literature

**Abstract:**

Chronic lymphocytic leukemia (CLL) exhibits substantial variability in disease course. The mutational status of the B-cell receptor immunoglobulin heavy variable (IGHV) chain is a critical prognostic factor, categorizing patients into mutated (M-IGHV) and unmutated (U-IGHV) groups. Recently, a third subgroup with borderline mutational status (BL-IGHV) has been identified, comprising approximately 5% of CLL cases. This study retrospectively analyzes the outcomes of 30 BL-IGHV mutated patients among a cohort of 653 CLL patients, focusing on time to first treatment (TTFT) and overall survival (OS). BL-IGHV patients had a short TTFT similar to U-IGHV patients (median 30.2 vs. 34 months; *p* = 0.9). Conversely, the OS of BL-IGHV patients resembled M-IGHV patients (median NR vs. 258 months; *p* = 1). Despite a similar incidence in unfavorable prognostic factors, the TTFT was shorter compared to other published cohorts. However, striking similarities with other experiences suggest that BL-IGHV mutated patients share common biological characteristics, biased IGHV gene usage and BCR subset frequency. These findings also underscore the need for multicentric efforts aggregating data on BL-IGHV CLL in order to elucidate its disease course and optimize therapeutic approaches for this rare subgroup. Until then, predicting outcomes and optimal management of BL-IGHV CLL will remain challenging.

## 1. Introduction

Chronic lymphocytic leukemia (CLL) is a B-cell neoplasm that represents the most common leukemia in the Western world and whose treatment landscape saw enormous advancements in the last decade [1]. Despite a generally indolent disease course, patients affected by CLL display significant survival differences due to both disease- and patient-related clinical and biological factors [2,3]. Among the latter, CLL harbors frequent cytogenetical abnormalities that convey worse prognosis, such as del(17p), del(11q) and complex karyotype (defined by >3 cytogenetical abnormalities), while others like trisomy 12 and del(13q) are instead associated with a more favorable disease course [4,5]. Mutations in the *TP53* and *NOTCH1* genes, both associated with poor outcomes, are two other fundamental prognostic factors, with the former also influencing treatment selection [6,7,8].

Since its discovery, the B-cell receptor (BCR) immunoglobulin heavy variable (IGHV) mutational status has been another one of CLL’s most essential prognostic factors [2]. The IGHV chain represents the antigen-binding site on the BCR and undergoes the process of somatic hypermutation (SHM) in germinal centers, through which mutations are introduced in the IGHV gene sequence leading to increased antigen affinity [9]. Among CLL patients, we can identify two groups based on the degree of SHM undergone by the IGHV gene. Patients displaying a SHM rate >2%, and conversely a IGHV identity to the germline sequence of <98%, termed mutated (M-IGHV) patients, consistently show better outcomes than unmutated (U-IGHV) ones, which in turn present a SHM rate <2% (or an IGHV identity to the germline sequence >98%) [10,11]. This difference in prognosis is at least partly explained by different BCR signaling capabilities. Indeed, BCRs carrying M-IGHV chains retain a decreased capacity to activate downstream signaling pathways, thus leading to anergy, while in U-IGHV CLL the BCR is more capable of responding to antigen stimulation, thereby sustaining cell proliferation and disease progression [12,13,14,15].

Currently, IGHV mutational status is a consolidated prognostic factor in CLL and is widely employed in clinical practice [16,17,18]. M-IGHV patients have historically been the subgroup that could benefit the most from chemo-immunotherapy (CIT), possibly achieving early disease eradication after fludarabine–cyclophosphamide–rituximab therapy [19,20]. U-IGHV patients, meanwhile, consistently show worse outcomes when treated with CIT compared to targeted agents, which remain the preferred option in these patients [21,22,23,24,25,26].

The importance of BCR immunoglobulins in CLL was also underscored by the discovery that around 40% of CLL patients express distinct stereotyped amino-acid sequences within the complementarity determining region of the IGHV chain [27]. Thus, based on the specific sequence, CLL patients can be grouped into different subsets with unique characteristics. As an example, subset #2 patients present an aggressive clinical course, subset #4 patients present a more favorable one, and subset #8 patients present an increased risk of Richter transformation [28].

More recently, evidence has emerged regarding the existence of a third group of patients with an IGHV identity to the germline sequence of 97–97.99%, defined as “borderline” mutated (BL-IGHV) patients. These represent a small fraction of CLL (~5%) patients whose clinical outcome is still poorly defined [29]. While BL-IGHV patients have historically been included in the M-IGHV group due to initial reports describing a similar time to first treatment (TTFT) [30], this view has been challenged when further analyses on the BL-IGHV subgroup revealed enrichment in BCR stereotype subset #2 and #169 cases, but also in cases harboring the IGLV3-21-R110 mutation, all features predictive of an adverse prognosis [27,31,32,33]. The emergence of this evidence led the European Research Initiative on CLL (ERIC) to advise caution on the prognostic interpretation of a BL-IGHV mutational status in their more recent recommendations [34,35]. At the present time, no specific treatment strategy is recommended for this group of patients as evidence is lacking.

Thus far, the clinical outcome of BL-IGHV patients has been explored by two works. Firstly, Davis and colleagues demonstrated how patients with an IGHV identity from 97.00% to 98.99% show an intermediate TTFT and overall survival (OS) [36]. In a second work, Raponi and colleagues investigated the clinical course of BL-IGHV patients in two large cohorts. In this work, BL-IGHV patients were defined as having an IGHV identity from 97.00% to 97.99% and presented a TTFT similar to M-IGHV patients [30]. It is worth noting how these two studies used different cutoffs of IGHV identity in the germline sequence to define the BL-IGHV group, underscoring how IGHV mutational status is still defined based on arbitrary cutoffs. Indeed, different groups also tried to re-explore the optimal cutoffs to discern between M-IGHV and U-IGHV patients or analyzing IGHV mutational status as a continuous variable, but without establishing new cutoffs in clinical practice [37,38].

Given these discrepancies and the as yet poorly defined disease course of BL-IGHV CLL, we decided to retrospectively review the outcome of BL-IGHV patients in relation to M-IGHV and U-IGHV ones in a large cohort of CLL patients followed by our institution since the year 2000.

## 2. Patients and Methods

### 2.1. Patients and Outcome Measures

The baseline clinical and biological characteristics at diagnosis of 653 patients were retrieved from electronic medical records. Cytogenetical information regarding FISH results for del(11q), del(13q), trisomy 12 and del(17q), karyotype analysis, and molecular results for TP53 and NOTCH1 mutations were recorded at the time of diagnosis. IGHV mutational status was defined in accordance with the 2022 ERIC recommendations [35]. In greater detail, M-IGHV and U-IGHV status was defined as a percentage of IGHV identity to the germline sequence of ≤96.99% and ≥98.00%, respectively [39]. To define a BL-IGHV mutational status, the cutoffs of 97.00–97.99% were used, since they were both the same as those employed in the largest study on BL-IGHV CLL by Raponi et al. and also those recommended by ERIC [30,35]. TTFT was defined from the date of diagnosis until the initiation of CLL-directed treatment (event) or last known follow-up (censored), OS from the date of diagnosis until death (event) or last visit (censored). Progression-free survival, defined from the start of therapy until progression or death, was calculated for BL-IGHV mutated patients that underwent first-line therapy. The three groups defined by IGHV mutational status were compared regarding TTFT and OS. The prognostic impact of IGHV mutational status on TTFT and OS in relation to the other prognostic factors was analyzed by means of univariate and multivariate analysis. PFS was analyzed in BL-IGHV patients according to the type of therapy administered (CIT vs. targeted therapy). The study was approved by the local ethics committee and was conducted according to the Declaration of Helsinki.

### 2.2. FISH Analysis

FISH was performed on standard cytogenetic preparations obtained from peripheral blood using the Vysis CLL FISH probe set (Abbott, Abbott Park, IL, USA) in two distinct hybridizations as previously reported [40]. The probe panel included LSI p53/LSI ATM and LSI D13S319/LSI13q34/CEN12. Co-denaturation at 75 °C for 5 min in ThermoBrite (Abbott) was followed by 16–18 h of hybridization at 37 °C. Post-hybridization washes were performed at 73 °C in 0.4 × SSC/0.3% NP40 for 2 min followed by 1 min in 2 × SSC/0.1% NP-40 at room temperature. An AxioImager microscope (Zeiss, Jena, Germany) equipped with appropriate filters was used to analyze 300 interphase nuclei for each probe. The cut-off for positive values was 10% for the deletion of 11q22.3 (ATM), 17p13.1 (TP53), and D13S319 (13q14.3) loci and 5% for trisomy 12.

### 2.3. Karyotype Analysis

Karyotype analysis was performed on peripheral blood specimens after a 72h stimulation with 500 µM CpG ODN DSP30 mitogen (Roche, Risch, CH, Basel, Switzerland) + 20 U/mL interleukin-2 (IL-2) (Roche). After overnight exposure to 1 µg of KaryoMax^®^ Colcemid^®^ Solution (Life technologies Corporation, Grand Island, NY, USA), cells were harvested by adding a 0.075 M KCL hypotonic solution and incubated for 30 min at room temperature, followed by fixation with Carnoy’s solution. Wright’s stain was diluted in Söerensen’s Buffer (0.06 M/L Na2HPO4/0.06 M/l KH2PO4) for G-Banding analysis. The slides were analyzed using the Metafer automated acquisition system (MetaSystems s.r.l., Milan, Italy), and the karyotype was described after the analysis of at least 20 G-banded metaphases in accordance with international guidelines (ISCN 2020).

### 2.4. TP53 and NOTCH1 Mutation Analysis

The mutation hotspots in the *TP53* (exons 2-11 and splicing sites) and *NOTCH1* (exon 34 and splicing sites) genes were analyzed via PCR amplification and direct sequencing as previously described by Rossi et al. [41].

### 2.5. IGHV Mutational Status Analysis

RNA extraction was performed from 2 × 10^6^ B cells using the RNeasy Total RNA kit (Qiagen, Venlo, The Netherlands) and then reverse transcribed using the SuperScript Premplification System (Life Technologies Inc.) for first-strand cDNA synthesis. The VH gene family was assigned as previously described [42]. VH gene sequence analysis was performed by amplifying 5 µL of the obtained cDNA using the appropriate VH leader and CH primers, and the product was then sequenced by an automated genetic analyzer (3130 ABI Applied Biosystem, Forster City, CA, USA). All sequences were aligned to the IMGT/VQUEST tool and stereotyped subsets were determined with the ARResT tool 1.99.8.

### 2.6. Statistical Analysis

Survival functions were estimated by means of the Kaplan–Meier method and compared with the Log-rank test. Hazard ratios (HR) with 95% confidential intervals (95%CI) were estimated with univariate analysis according to the Cox proportional hazards model. The presence of statistically significant differences between groups was assessed by means of the Kruskal–Wallis test, chi-square test or Fisher’s exact test as appropriate. Univariate and multivariate analyses for clinical and biological variables were conducted using the Cox proportional hazards model. Variables that provided significant results in univariate analysis or whose distribution was significantly different among the three groups were included in the multivariate models.

## 3. Results

Baseline population characteristics for the three groups are reported in Table 1. BL-IGHV mutated patients more frequently presented with an advanced stage at diagnosis compared to M-IGHV ones (*p* = 0.002 for RAI stages II–IV and *p* < 0.001 for BINET stages B-C) and more frequently harbored del(11q) (10.7% vs. 2.4%, *p* = 0.03) like the U-IGHV group (10.7% vs. 21.7%; *p* = 0.56). Only a single BL-IGHV mutated patient presented trisomy 12, whereas it was observed in 10% and 13.6% of M-IGHV and U-IGHV patients, respectively. Del(13q) was present at similar rates as in M-IGHV (57.1% vs. 68.2%; *p* = 0.55) and at lower, although not statistically significantly, rates than in U-IGHV (57.1% vs. 42.2%; *p* = 0.16). The rates of del(17p) and TP53 mutations were instead more similar to those found in M-IGHV patients and numerically lower when compared to U-IGHV patients (0% vs. 14.3%, *p* = 0.05 for del(17p) and 6.7% vs. 22.9%, *p* = 0.19 for *TP53*). The same was also true for *NOTCH1* mutations (6.7% vs. 22.1% in U-IGHV, *p* = 0.3). Patients with a BL-IGHV mutational status frequently harbored chromosomal abnormalities, with only 17.6% of patients having a normal karyotype. Moreover, BL-IGHV patients had the highest rate of complex karyotypes, although without a statistically significant difference when compared to the other groups. Regarding the choice of first line therapy, this was almost superimposable between BL-IGHV and U-IGHV patients, whereas a larger fraction of M-IGHV patients received CIT (Table 1).

Interestingly, the BL-IGHV group was particularly enriched in cases carrying *IGHV 2-5* (6.7% vs. 2.9% and 0.4% for M-IGHV and U-IGHV, respectively; *p* = 0.01), *IGHV 3-21* (23.3% vs. 3.7% and 3.7%; *p* < 0.01), *IGHV 3-23* (16.7% vs. 10.9% and 3.7%; *p* < 0.01%), and *IGHV 3-74* (13.3% vs. 3.1% and 1.1%; *p* < 0.01) genes (Appendix A). As reported in previous studies [36], the BL-IGHV group was enriched in patients belonging to the BCR subset #2 (10% vs. 2.6% and 0.7%; *p* = 0.01 and *p* = 0.002 compared with M-IGHV and U-IGHV, respectively) (Appendix A).

Patients with a BL-IGHV mutational status had a significantly shorter TTFT compared to M-IGHV (median 30.2 vs. 212.6 months; HR: 3.84; 95% CI: 2.43–6.07; *p* < 0.001), which more closely resembled that of U-IGHV patients (median 30.2 vs. 34 months; HR: 0.97; 95% CI: 0.62–1.50; *p* = 0.9) (Figure 1A). These results held true when restricting the analysis to patients with RAI stages 0-I (median 13.7 vs. 229.7 months for BL-IGHV vs. M-IGHV; HR: 7.27; 95% CI: 3.90–13.58; *p* < 0.001), BINET stage A (median 14.3 vs. 229.7 months; HR: 6.02; 95% CI: 3.25–11.14; *p* < 0.001), or when removing subset #2 patients from the analysis (median 30.2 vs. 221.7 months; HR: 3.90; 95% CI: 2.42–6.47; *p* < 0.001). Conversely, the OS of BL-IGHV patients was similar to that of M-IGHV ones (median NR vs. 258 months; HR: 1.01; 95% CI: 0.41–2.48; *p* = 1) and was significantly longer than that of the U-IGHV group (median NR vs. 130 months; HR: 0.35; 95% CI: 0.14–0.85; *p* = 0.006) (Figure 1B), possibly reflecting a favorable response to therapies in individuals with a BL-IGHV status. In both univariate and multivariate analysis, a BL-IGHV mutational status was an independent predictor of a shorter TTFT, but not OS (Appendix A). Among the 22 BL-IGHV mutated patients who underwent treatment, 5 (22.7%) were treated with targeted agents (2 with ibrutinib, 2 with ibrutinib + venetoclax, and 1 with rituximab + venetoclax) and 18 (78.3%) with CIT. No statistically significant differences were observed in PFS between those treated with targeted agents and CIT (median 57.7 vs. 48.1 months; HR 0.39; 95% CI: 0.05–3.08; *p* = 0.14).

## 4. Discussion

CLL with a BL-IGHV mutational status emerges as a subgroup of CLL characterized by clinical and biological characteristics shared at times with M-IGHV and at others with U-IGHV CLL, but that collectively compose a unique landscape. Similarly to U-IGHV patients, BL-IGHV patients had an high rate of del(11q), which is coherent with their higher Rai and Binet stage at diagnosis when compared with M-IGHV patients, as the presence of del(11q) is associated with lymphadenopathy, a higher clinical stage, and faster disease progression [43,44]. Conversely, the BL-IGHV group shared similarly low rates of unfavorable prognostic features with the M-IGHV group such as del(17p), *TP53* and *NOTCH1* mutations. A unique feature was instead the very low rate of trisomy 12, which is normally observed in almost 15% of CLL patients [5], but appeared at very low rates in BL-IGHV CLL.

While these numbers must be interpreted with caution given the low number of patients with a BL-IGHV mutational status included in this study, the mixture of both favorable and unfavorable features is coherent with the observed favorable OS, despite the rapid disease kinetics demonstrated by the short TTFT.

The biased usage of specific IGHV genes in BL-IGHV CLL was already described by Davis et al., who reported a high fraction of BL-IGHV cases using the *IGHV3-23* gene [36]. To add to this knowledge, in this study, we also reported on the biased use of *IGHV2-5, IGHV3-21* and *IGHV3-74* genes in BL-IGHV CLL. Unfortunately, Raponi et al. did not report on the frequency of IGHV gene usage in their work [30]. It is to be noted that IGHV gene usage and BCR subtype frequency may vary among different populations due to the contribution of antigen selection in CLL, and as such prevalent IGHV genes in BL-IGHV CLL could also vary among different regions [45].

In the seminal work by Davis et al. that for the first time analyzed the characteristics and outcomes of BL-IGHV patients, these patients were found to have an intermediate TTFT and OS compared with the other two subgroups. While the frequency of cytogenetical abnormalities for each group was not reported, as described above, their results are in accordance with ours regarding IGHV gene usage but also the enrichment in subset #2 cases [36]. A caveat of this study that hinders comparison with ours, however, is a different design with the choice of the 97–98.99% cutoffs to define BL-IGHV mutated patients. Indeed, including a fraction of patients more commonly classified as having a U-IGHV mutational status in the BL-IGHV makes comparison of time-to-event outcomes particularly challenging and unreliable. Yet, this also highlights the fact that the BL-IGHV group is defined based on arbitrary cutoffs, and more studies that correlate a BL-IGHV mutational status with specific biological subgroups of CLL are needed.

Compared to the other experience with BL-IGHV mutated patients reported in the literature, our data show both differences and similarities in relation to those published by Raponi and colleagues in 2020. In accordance with our results, they also observed high rates of del(11q), subset #2 and low rates of *NOTCH1* mutations in BL-IGHV, while only the frequencies of *TP53* mutations and del(17p) were higher than what we observed [30]. However, the most striking difference was observed in TTFT, which was particularly favorable, unlike in our cohort. These differences are hard to interpret, as there are most likely derived from bias induced by the low number of BL-IGHV patients included in both analyses. Another source of possible bias is the fact that the retrospective measurement of TTFT is influenced by the time at which patients come to medical attention and are diagnosed with CLL, which is a chronic disease that can persist undiagnosed for years, thus shortening TTFT.

What has emerged thus far is that BL-IGHV patients present a heterogeneous disease course, which may vary depending on the specific series and associated prognostic factors. As already mentioned, a possible explanation for this heterogeneity is the overall low prevalence of BL-IGHV mutated patients in CLL cohorts, which ranges from 4.3 to 7.5% [30,36], hindering a more accurate prediction of time-to-event outcomes. To overcome these limitations and dilute bias, studies aggregating large numbers of BL-IGHV are needed, which could possibly resolve the issue especially regarding TTFT compared to M-IGHV and U-IGHV patients. Another important determinant of prognosis in BL-IGHV CLL is the presence of the newly recognized IGLV3-21R110 mutation, consistently associated with poor outcomes [31,32]. The presence of the IGLV3-21R110 mutation has not been assessed in our or in previous works, but differences in its frequency among different study populations may also be responsible for inconsistent results.

Despite these inconsistencies, the identification of a BL-IGHV mutated group in CLL is not unjustified. Indeed, from our and previous studies, the association of different biological predictors with a BL-IGHV mutational status starts to emerge, depicting a subgroup of CLL frequently harboring del(11q), low rates of *NOTCH1* mutations, biased usage of specific IGHV genes and an enrichment in subset #2 cases. Moreover, DNA methylation studies recently identified a distinct clinico-biological CLL subgroup, termed intermediate-CLL, characterized by moderately mutated IGHV genes (95–98% identity to the germline sequence), enrichment in subset #2 cases, SF3B1 mutations, and an intermediate TTFT with relatively favorable OS [46,47,48]. Intermediate-CLL finds its postulated cellular origin in a germinal center-experienced B-cell that has undergone early selection through the germinal center reaction, thus retaining both an intermediate methylation program and a somatic hypermutation burden that falls in the BL-IGHV range [47]. While awaiting confirmatory studies, these findings could represent the first seminal evidence suggesting that BL-IGHV mutated CLL might have superimposable biological characteristics to intermediate methylated CLL.

Another interesting question is how this subgroup of patients responds to different therapies, which was never assessed in previous studies. In our cohort, most BL-IGHV patients received first line therapy with CIT and only a minority with targeted agents, reflecting a clinical practice where BL-IGHV were treated in a similar manner as M-IGHV ones. Due to the low number of patients included in this exploratory analysis and the short follow-up of BL-IGHV patients treated with targeted agents, we could not demonstrate a significant difference in PFS. Nevertheless, it appears that BL-IGHV patients still achieved favorable PFS rates with CIT, much like M-IGHV patients, which could also explain their favorable OS. Again, further multicentric efforts will help to clarify this question in the future.

## 5. Conclusions

While unable to resolve the outcome of BL-IGHV mutated patients, this study adds to the emerging evidence describing CLL with a BL-IGHV mutational status as a distinct subgroup of a heterogeneous disease, characterized by a higher frequency of specific cytogenetical abnormalities, biased IGHV gene usage, enrichment in BCR subset #2 cases, and thus very heterogeneous outcomes. To address the limitations posed by the low prevalence of BL-IGHV CLL patients, to clarify their disease course, and define whether a specific therapeutic approach is warranted for this rare subgroup of CLL, the aggregation of data from large numbers of BL-IGHV mutated patients through multicentric efforts is needed. Until then, the prognosis and the optimal therapeutic approach to BL-IGHV CLL will remain elusive.

## Figures and Tables

**Figure 1 cancers-16-01095-f001:**
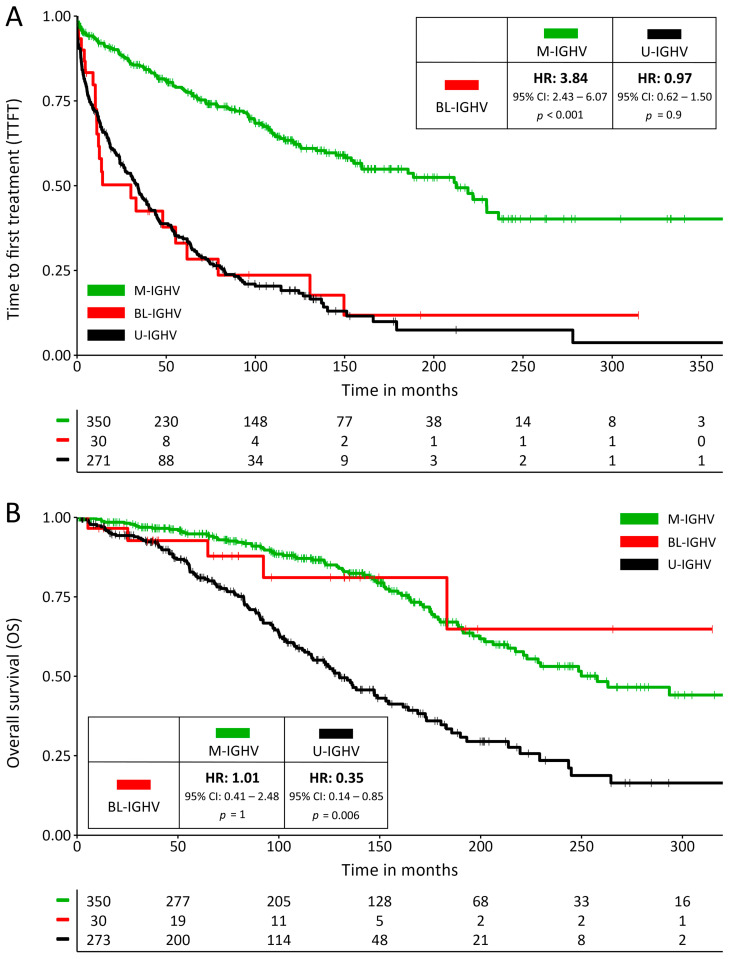
(**A**) Survival curves for time to first treatment (TTFT) and (**B**) for overall survival; the inserts show hazard ratios and their confidence intervals for BL-IGHV patients compared to M-IGHV and U-IGHV patients.

**Table 1 cancers-16-01095-t001:** Baseline population characteristics at diagnosis.

	M-IGHV (n = 350; 53.6%)	BL-IGHV (n = 30; 4.59%)	U-IGHV (n = 273; 41.8%)	*p* Values ^a^
**Sex (%)**				0.33
F	139 (39.7)	15 (50)	100 (36.6)	
M	211 (60.3)	15 (50)	173 (63.4)	
**Median age, years (IQR)**				0.26
	62.9 (14.8)	66.4 (16)	65.2 (16.6)	
**RAI stage (%)**				**<0.001**
0-I	284 (86.6)	16 (61.5)	182 (69.2)	
II	32 (9.8)	7 (26.9)	63 (24)	
III	9 (2.7)	1 (3.9)	3 (1.1)	
IV	3 (0.9)	2 (7.7)	15 (5.7)	
Not reported	22 (6.3)	4 (13.3)	10 (3.7)	
**BINET stage (%)**				**<0.001**
A	293 (90.4)	16 (61.5)	170 (67.2)	
B	20 (6.2)	7 (26.9)	65 (25.7)	
C	11 (3.4)	3 (11.5)	18 (7.1)	
Not reported	26 (7.4)	4 (13.3)	20 (7.3)	
**FISH (%)**				
del(11q)	7 (2.4)	3 (10.7)	50 (21.7)	**<0.001**
del(13q)	182 (62.8)	16 (57.1)	97 (42.2)	**<0.001**
+12	35 (12.1)	1 (3.6)	37 (16.1)	0.12
Del(17p)	12 (4.1)	0 (0.0)	33 (14.3)	**<0.001**
Not performed	60 (17.1)	2 (6.7)	43 (15.8)	
**TP53 (%)**				**0.01**
Mutated	7 (8.43)	1 (6.7)	27 (22.9)	
Wild type	76 (91.6)	14 (93.3)	91 (77.1)	
Not performed	267 (76.3)	15 (50)	155 (56.8)	
**NOTCH 1 (%)**				**0.01**
Mutated	7 (8)	1 (6.7)	25 (22.1)	
Wild type	81 (92)	14 (93.3)	88 (77.9)	
Not performed	262 (74.9)	15 (50)	160 (58.6)	
**Karyotype (%)**				**<0.001**
Normal	63 (50.8)	3 (17.6)	32 (22.5)	
<2 abnormalities	41 (33.1)	9 (52.9)	70 (49.3)	
CK	13 (10.5)	4 (23.5)	15 (10.6)	
High-CK	7 (5.7)	1 (5.9)	25 (17.6)	
Not performed	226	13	131	
**IGHV subset #2**	9 (2.6)	4 (12.9)	2 (0.7)	**0.003**
**First line therapy** **(%)**				**0.038**
CIT	112 (91.1)	18 (81.8)	171 (81)	
Targeted	11 (8.9)	5 (16.7)	40 (19)	

CK: Complex karyotype (≥3 cytogenetical abnormalities); High-CK: High complex karyotype (≥5 cytogenetical abnormalities). All percentages are calculated excluding patients with missing data from the total count. ^a^ Overall *p*-values between the three groups, relevant pair-wise *p*-values reported in text; *p*-values below the significance level are indicated in bold.

## Data Availability

The datasets generated and analyzed during the current study are not publicly available due to data protection and lack of consent from the patients. Access to data is strictly limited to the researchers who have obtained permission for data processing.

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
