# Peer review of "Chronic Lymphocytic Leukemia (CLL) with Borderline Immunoglobulin Heavy Chain Mutational Status, a Rare Subgroup of CLL with Variable Disease Course"

_cancers, 2024, doi:10.3390/cancers16061095_

Round 1

Reviewer 1 Report

Comments and Suggestions for Authors

Dear Editor,

Dear authors,

In this article, Angotzi et al. present a retrospective study on Borderline (BL) -IGHV mutational status in Chronic Lymphocytic leukemia (CLL). This entity is rare, and particularly there is a lack of data on this subgroup of CLL in the published medical literature. The current paper presents interesting data and can be considered as a stepping stone for future large cohort analysis to better understand the prognosis of BL-IGHV. Please find below minor comments and suggestions:

-          The introduction is very brief. I suggest adding more details about CLL. And also, about other prognostic factors that Include cytogenetic anomalies (del17p, del11q, +12 and del13q) (PMID: 33091559, 29540348). Especially that karyotype and FISH results, as well as some molecular analysis, are reported in the Results section.

-          And I guess that it should be clearly stated that the prognosis will be defined according to the IGHV mutational status, along with the presence of other cytogenomic changes.

-          I suggest also, introducing CLL subsets and their importance in the introduction section, since subset #2 was investigated and reported in the results section (33131249).

-          Is there any enrichment in any IGHV(1-6) gene among BL-IGHV patients? And what about the other subsets, other than #2? I suggest adding a table contain this information.

-          Concerning the treatment: in the table1 it is stated that 4 BL-IGHV patients had targeted therapies, however in the text line 168 it is said that 5 BL-IGHV patients had targeted therapies, and yet between brackets: 2+2+2=6. Which number is correct 4, or 5, or 6? Please explain or correct.

-          In the discussion section, I suggest adding any idea about the differences in CLL among populations, the IGHV gene distribution, also subsets distributions, and even the fraction of patients stereotyped (into subsets #) are different among populations (PMID: 37795707). This will support the idea of the heterogeneity of CLL (line 146-147).

Comments on the Quality of English Language

No comment.

Author Response

Dear reviewer,

We kindly thank you for your timely and thorough review and greatly appreciated your comments. We have modified our work in order to comply with yours and other reviewer’s suggestions. Please find our point-by-point responses as follows:

-          The introduction is very brief. I suggest adding more details about CLL. And also, about other prognostic factors that Include cytogenetic anomalies (del17p, del11q, +12 and del13q) (PMID: 33091559, 29540348). Especially that karyotype and FISH results, as well as some molecular analysis, are reported in the Results section.

  • The introduction was revisited, specific details regarding cytogenetic anomalies (karyotype and FISH results) and molecular abnormalities were added (lines 43-53).

-          And I guess that it should be clearly stated that the prognosis will be defined according to the IGHV mutational status, along with the presence of other cytogenomic changes.

  • Details regarding the definition of prognosis were added in the patients and methods section in lines 129-131 (“The prognostic impact of IGHV mutational status on TTFT and OS in relation to the other prognostic factors was analyzed by univariate and multivariate analysis”)

-          I suggest also, introducing CLL subsets and their importance in the introduction section, since subset #2 was investigated and reported in the results section (33131249).

  • Information regarding CLL subsets and their importance was added in the introduction in the paragraph in lines 76 – 81.

-          Is there any enrichment in any IGHV(1-6) gene among BL-IGHV patients? And what about the other subsets, other than #2? I suggest adding a table contain this information.

  • As suggested, we thoroughly added information regarding IGHV gene usage and subtypes in the results section (lines 210-216). The complete list of IGHV gene usage in our cohort and CLL subsets identified was included in two additional supplementary tables (tables S5 and table S6).

-          Concerning the treatment: in the table1 it is stated that 4 BL-IGHV patients had targeted therapies, however in the text line 168 it is said that 5 BL-IGHV patients had targeted therapies, and yet between brackets: 2+2+2=6. Which number is correct 4, or 5, or 6? Please explain or correct.

  • The correct number of BL-IGHV patients treated with targeted agents (5) was inserted in table 1 and corrected in the text (line 235 - 236).

 -          In the discussion section, I suggest adding any idea about the differences in CLL among populations, the IGHV gene distribution, also subsets distributions, and even the fraction of patients stereotyped (into subsets #) are different among populations (PMID: 37795707). This will support the idea of the heterogeneity of CLL (line 146-147).

  • We commented on the notion of difference in IGHV gene distribution and subsets among different populations in lines 268 – 275.

Reviewer 2 Report

Comments and Suggestions for Authors

This is a well-written manuscript designed in a strict style and well compressed for the "Communication" article. Despite the study population size is quite small (only 30 persons), I accept the main idea streamed by the authors in the paper. It seems that the minor group of BL-IGHV patients requires neat approach and tailored treatment options. The authors proposed 97.00 – 97.99% cut-off albeit the cut-off for CLL is yet debatable issue. For sometime now, plenty of researchers and clinicians have lamented the fact that accuracy of cut-off criteria requires corrections while variety of combination of mutations sometimes remains overlooked. It seems on the surface that the 98% cut-off is the most appropriate and acceptable over the world for the OS prediction and selection of treatment options. Recent study (PMID 31849198) purposed to observe differences in a large population (on 595 subjects with CLL) within 95%-99% identity by splitting the study population on subgroups in a 1% step wise manner. Hoowever no significant differences for TTFT and OS within groups with >98% identity (98-98.99,  99-99.99 etc.) and within those subgroups with <98% identity (95-95.99%, 96-96.99% etc) were established. 

Although I have to admit that ethnic features and national criteria and guideline for diagnosis might had great impact in the final results and assumptions. Moreover, other researcher group including PMID 3137122 assessed mutational status and loading in a retrospective study, and eventually articulated the correction of 98% cut-off as the most appropriate to determine progression-free survival considering two treatment strategies in the studied population. Thus is might seems that cut-off status beyond the 98% is obscure.

Yet, as far-fetched as that seems, I have to accept the general idea of the authors that 97-97.99% cut-of, or so-called BL-IGHV status, merits attentions since there are various idea to establish correct cut-off including that is to suggest that IGHV deviation is a continuous variable with the significant association to overall survival and progression-free survival. That approahc might be the closest to the proposed idea abut borderline mutational status declared by the authors. So I suggest that the given information in a Communication format would be successful for the further data collection in order to elaborate better therapy strategy and prediction model.

Besides, we have to consider about specifically affected patients with 97-97.99% mutational status despite they comprises about 5% of the totally examined CLL patients population, which is rather large population and affordable to be considered in a separate way. 

Author Response

Dear reviewer,

Thank you for your timely and useful feedback on our work. In order to properly address the other reviewer’s concerns, we had to expand the introduction, methods and discussion sections.

  • We mentioned the issue regarding the optimal IGHV cutoff in lines 103 – 108.

  • To further clarify on the reason for choosing the 97-97.99% cutoff to define BL-IGHV CLL, we added lines 121 – 124 in the patients and methods section.

Reviewer 3 Report

Comments and Suggestions for Authors

The authors carried out a retrospective study on a cohort of 653 patients focusing on TTFT and OS relating not only to U-CLL and M-CLL cases but also those cases defined as borderline.

I find that the work is well written and presents a clear message: BL-IGHV patients have an OS similar to mutated cases while they have a shorter TTFT than Unmutated patients.

It is important to continue with molecular studies aimed at a better biological definition of cases defined as borderline.

Comments on the Quality of English Language

I recommend a better spelling check

Author Response

Dear reviewer,

Thank you for you timely review. In order to accommodate the other reviewer’s comments, we extended different sections of the manuscript. As you kindly recommended, we also performed a thorough spelling check.

Reviewer 4 Report

Comments and Suggestions for Authors

Dear Authors,

After carefully reviewing the manuscript, I found several areas that need significant improvement before it can be considered for publication in our journal. Below are my comments and suggestions for revision:

1.       Definition of BL-IGHV: The criteria for defining BL-IGHV mutational status as IGHV gene identity to the germ line sequence of 97-97.99% lacks sufficient clarity. Given the different prognostic conclusions reached by Raponi et al., 2020 and Davis et al., 2016 based on varying percentage thresholds, it's imperative that you elaborate on your chosen criteria in detail. Additionally, the ambiguity surrounding BL-IGHV as a prognostic criterion for CLL warrants thorough discussion in both the introduction and discussion sections.

2.       Comparison with Raponi et al., 2020: The similarities between your study and Raponi et al., 2020 are noted, but the conclusions drawn in your manuscript appear weaker. I recommend discussing the findings of Raponi et al. in greater detail to contextualize your own results.

3.       Introduction: The introduction section is too brief and lacks essential details. Please expand on topics such as BL-IGHV, the significance of IGHV in the context of BCR, and the current prognosis and treatment strategies for this subset of CLL patients.

4.       Materials and Methods: This section requires more thorough explanation. Subheadings for karyotyping, FISH analysis, TP53 mutational analysis, and the method for IGHV mutation analysis should be included and described in detail.

5.       Results:

a.       Small Sample Size: With only 30 BL-IGHV patients out of 653 (4%), the conclusions drawn may be subject to bias. Please address this limitation.

b.       Clarification Needed: The term "whole population" in line 81 requires clarification. Please specify whether it refers to the entire cohort and make necessary corrections.

c.       Significance of Chromosomal Abnormalities: Please elucidate the significance of chromosomal abnormalities like del(11q) in relation to BL-IGHV.

d.       Contradictory TTFT and OS Data: The TTFT and OS data presented appear contradictory and differ from previous studies such as Davis et al., 2016 and Raponi et al., 2020. A more thorough justification and discussion are needed to resolve this discrepancy.

6.       Discussion: The discussion section is brief and lacks adequate explanation. Please expand on your findings and their implications, addressing the points raised in the results section.

Comments on the Quality of English Language

English is fine.

Author Response

Dear reviewer,

We kindly thank you for your interesting and extensive comments, please find below our response to you comments and suggestions:

  1. Definition of BL-IGHV: The criteria for defining BL-IGHV mutational status as IGHV gene identity to the germ line sequence of 97-97.99% lacks sufficient clarity. Given the different prognostic conclusions reached by Raponi et al., 2020 and Davis et al., 2016 based on varying percentage thresholds, it's imperative that you elaborate on your chosen criteria in detail. Additionally, the ambiguity surrounding BL-IGHV as a prognostic criterion for CLL warrants thorough discussion in both the introduction and discussion sections.

- The criteria for defining BL-IGHV mutational status were added and clarified in the patients and methods section in lines 121 – 124. The issue regarding different percentage thresholds and their impact on the interpretation of results was addressed both in the introduction and discussion sections in lines 101 – 106 and lines 280 – 287. The ambiguity surrounding BL-IGHV as a prognostic factor was further addressed and expanded in lines 86 – 94.

  1. Comparison with Raponi et al., 2020: The similarities between your study and Raponi et al., 2020 are noted, but the conclusions drawn in your manuscript appear weaker. I recommend discussing the findings of Raponi et al. in greater detail to contextualize your own results.

- The comparison between our study and Raponi et al. was expanded in the discussion section in lines 276 – 287.

  1. Introduction: The introduction section is too brief and lacks essential details. Please expand on topics such as BL-IGHV, the significance of IGHV in the context of BCR, and the current prognosis and treatment strategies for this subset of CLL patients.

- The introduction section was greatly expanded to include more details. The topic of BL-IGHV is discussed in lines 82 – 94; the significance of IGHV in the context of BCR was discussed in lines 64 – 68; current treatment strategies in relation to IGHV mutational status are underlined in lines 69 – 75.

  1. Materials and Methods: This section requires more thorough explanation. Subheadings for karyotyping, FISH analysis, TP53 mutational analysis, and the method for IGHV mutation analysis should be included and described in detail.

- Detailed explanation regarding methods for karyotyping (lines 155 – 164), FISH analysis (lines 143 – 153), TP53 and NOTCH1 analysis (lines 165 – 168) and IGHV mutational status determination (lines 169 – 177) were added in different paragraphs with subheadings.

  1. Results:

5a. Small Sample Size: With only 30 BL-IGHV patients out of 653 (4%), the conclusions drawn may be subject to bias. Please address this limitation.

- The limitations of small sample size were addressed in lines 264 – 267, lines 294 – 296 and lines 311 – 314.

5B. Clarification Needed: The term "whole population" in line 81 requires clarification. Please specify whether it refers to the entire cohort and make necessary corrections.

- The term “whole population” wash changed to “Baseline population characteristics for the three groups” now un line 191 – 192.

5C. Significance of Chromosomal Abnormalities: Please elucidate the significance of chromosomal abnormalities like del(11q) in relation to BL-IGHV.

- The significance of chromosomal abnormalities in relation to BL-IGHV was underlined in lines 255 – 259.

5D. Contradictory TTFT and OS Data: The TTFT and OS data presented appear contradictory and differ from previous studies such as Davis et al., 2016 and Raponi et al., 2020. A more thorough justification and discussion are needed to resolve this discrepancy.

- We expanded on the interpretation of TTFT and OS given their discrepancy with previous studies in lines 280 – 284 and lines 294 – 299.

  1. Discussion: The discussion section is brief and lacks adequate explanation. Please expand on your findings and their implications, addressing the points raised in the results section.

- The discussion was extended to include explanations on the aforementioned point in lines 253 – 287, lines 289 – 299, lines 311 – 314, lines 320 – 323 and lines 335 – 343.

Round 2

Reviewer 4 Report

Comments and Suggestions for Authors

No comments